# Energy-Efficient UAV-Enabled MEC System: Bits Allocation Optimization and Trajectory Design

**DOI:** 10.3390/s19204521

**Published:** 2019-10-17

**Authors:** Linpei Li, Xiangming Wen, Zhaoming Lu, Qi Pan, Wenpeng Jing, Zhiqun Hu

**Affiliations:** 1School of Information and Communication Engineering, Beijing University of Posts and Telecommunications, Beijing 100876, China; llinp1993@163.com (L.L.); xiangmw@bupt.edu.cn (X.W.); panqiouc@sina.com (Q.P.); jingwenpeng@bupt.edu.cn (W.J.); 2Beijing Key Laboratory of Network System Architecture and Convergence, Beijing University of Posts and Telecommunications, Beijing 100876, China; 3Beijing Laboratory of Advanced Information Networks, Beijing University of Posts and Telecommunications, Beijing 100876, China; 4School of Computing and Information Engineering, Hubei University, Wuhan 430062, China; zhiqunhu520@163.com

**Keywords:** wireless communication, unmanned aerial vehicles, mobile computing, mobile edge computing, offloading, computation, bits allocation, trajectory design

## Abstract

The unmanned aerial vehicle (UAV) enabled mobile edge computing (MEC) system is attracting a lot of attentions for the potential of low latency and low transmission energy consumption, due to the advantages of high mobility and easy deployment. It has been widely applied to provide communication and computing services, especially in Internet of Things (IoT). However, there are still some challenges in the UAV-enabled MEC system. Firstly, the endurance of the UAV is limited and further impacts the performance of the system. Secondly, mobile devices are battery-powered and the batteries of some devices are hard to change. Therefore, in this paper, a UAV-enabled MEC system in which the UAV is empowered to have computing capability and provides tasks offloading service is studied. The total energy consumption of the UAV-enabled system, which includes the energy consumption of the UAV and the energy consumption of the ground users, is minimized under the constraints of the UAV’s energy budget, the number of each task’s bits, the causality of the data and the velocity of the UAV. The bits allocation of uploading data, computing data, downloading data and the trajectory of the UAV are jointly optimized with the goal of minimizing the total energy consumption. Moreover, a two-stage alternating algorithm is proposed to solve the non-convex formulated problem. Finally, the simulation results show the superiority of the proposed scheme compared with other benchmark schemes. Finally, the performance of the proposed scheme is demonstrated under different settings.

## 1. Introduction

With the technological evolution of user devices, the computation-intensive and latency-critical applications, such as virtual reality, face recognition and Industrial Internet of Things (IIoT), have become more and more popular. However, the requirements of high quality communications have simultaneously posed big pressure on the user devices [1,2]. Although the user devices are equipped with powerful central processing unit (CPU), they are not able to deal with the ever-increasing amount of data. Mobile cloud computing (MCC) gives the chance to overcome the computing difficulties of users, which can offload some computation tasks to the resource-rich cloud infrastructures and relieve the computing pressure for users [1].

The latency and the energy consumption of the user devices are reduced owing to the popularity of MCC. Nevertheless, the cloud infrastructures are not always located nearby the user devices, which causes the huge additional load and long latency [2]. Then, the concept of mobile edge computing (MEC) is introduced. MEC offers the capacity of storage and processing by deploying cloud resources in proximity to the user devices [3]. In the MEC framework, mobile devices can offload their tasks to the MEC servers on the edge of the network, rather than utilizing the core network [3,4]. MEC brings the opportunity to reduce the energy consumption and the long latency caused by the long-distance transmission between user devices and remote servers.

MEC enables the mobile base station (BS) to be equipped with powerful computing capability, which significantly relieves the pressure of mobile networks. However, in the areas where the communication facilities are sparsely distributed, especially in IoT, the fixed MEC server can not satisfy the computation requirements of the remotes users. Besides, when communication infrastructures are damaged, malfunctional or overloaded, the computing tasks of the users can not be processed in time. Hence, it is necessary to devise a scheme that the MEC servers can satisfy the computation demand in a timely fashion. Fortunately, UAVs have the characteristics of high mobility and easy deployment, making the on-demand communication services provision possible [5,6]. Recently, UAVs have attracted a lot of attention in many applications, such as delivery, farming, rescue response and communication services [7]. UAVs are expected to play the role of wireless communication platforms equipped with communication modules [8]. An UAV-enabled MEC system can provide on-demand computation services for the mobile users with on-demand mobility compared with the fixed communication infrastructures. In the areas where the fixed communication infrastructures can not satisfy the computing requirements of ground users, such as the places which are remote from the communication facilities and the places destroyed by the natural disasters, the advantages of UAV-enable MEC system are highlighted. The UAV is capable of flying to the specific areas to help the users with computational requirements, e.g., monitoring devices, to compute the tasks. Owing to the on-demand mobility, the UAV-enabled MEC system can greatly relieve the load of computation in the specific areas.

Despite the advantages UAV-enabled MEC systems bring, there are still some challenges. Among these challenges, the most typical and urgent one is the energy-efficient problem. UAV’s time of endurance is finite due to the limitation of the battery technology [9]. The duration of flight is also affected due to the payload and the energy consumption of the communication and computation in a UAV-enabled MEC system. Moreover, the mobile devices are always battery-powered, which compromises the performance of offloading owing to the long-distance transmission. It is difficult to change the batteries of the mobile devices in some areas which are hard to reach, e.g., monitoring devices in IoT [10,11]. Therefore, it is necessary to design an energy-efficient UAV-assisted MEC system in order to minimize the energy consumption of both the UAV and mobile devices.

### 1.1. Related Work

The first concept of edge computing is proposed in 2009, which is named cloudlet [2,12]. There  has been a lot of research about MEC in recent years, concerning the energy consumption, latency and so on. In [4], the authors designed an energy-efficient MEC offloading mechanism for mobile devices in 5G heterogeneous networks. The mechanism aimed at minimizing the system’s energy consumption and ensuring the latency constraints of the computation task. Authors in [1] proposed an algorithm aiming at solving the minimum energy consumption problem in deadline-aware MEC system. The mobility of the mobile devices was also considered in the deadline-aware MEC system. In addition to the energy-efficient mechanisms, the latency is also investigated. In [10], an effective dynamic computation offloading scheme for energy harvesting mobile devices was proposed, aiming at minimizing the execution cost which consists of execution delay and task failure.

Although MEC reduces the energy consumption and latency compared with MCC, it is difficult to satisfy the communication requirements of mobile users in some specific areas, such as hotspot areas or the areas where the communication facilities are distributed sparsely. UAV is recognized to be one of promising wireless communication platforms due to the advantages of high mobility and easy deployment [13]. UAVs, acting as flying base stations or relays, have attracted people’s attention in recent years. The placement of the UAV to maximize the coverage has been studied in a lot of research. In [14], the optimal altitude of the UAV-based base station was analyzed for the maximal communication coverage. In [15], an efficient UAV 3D placement with the purpose of maximizing the covered users based on the optimal altitude was proposed. In [16], the authors studied a novel 3D UAV placement with the objective of maximizing the number of covered users according to different requirements of quality of service (QoS). Besides, the energy consumption of both the UAV and mobile users have also been taken into consideration. In [17], the propulsion energy consumption model of the fixed-wing UAV was derived and an efficient trajectory maximizing the UAV’s energy efficiency was designed. In [11], given the locations of active devices, the optimal locations of the UAVs and the associations with devices were determined with the objective of minimizing the transmission power of devices. Moreover, the resource management of the UAV-aided communications has also been investigated. In [18], an energy efficient solution in UAV-supported multi-level architecture is employed by grouping uer equipments in clusters based on reinforcement learning. In [19], based on the above cluster solution, the authors proposed a framework that combined UAV-support with wireless powered communication (WPC) techniques to further improve energy efficiency in a distributed non-orthogonal multiple access (NOMA) public safety networks (PSN). The optimal position of the UAV in the Euclidean 3D space was determined by maximizing the coalition head’s total energy availability. In [20], a game theoretic framework for load balancing between LTE-Unlicensed unmanned aerial base stations (UABSs) and WiFi access points (APs) based on the loads at the UABSs and the ground APs was proposed. In [21], a framework for the coverage and rate analysis was derived, considering the coexistence between the UAV as a flying base station, and an underlaid device-to-device (D2D) communication network. The framework considered two scenarios: a static UAV and a mobile UAV. The overall outage probabiliy of the D2D users was derived considering multiple retransmissions for the UAV and D2D users.

UAVs endowed with computing capabilities have become more and more popular in recent years, owing to the on-demand communication services provision and flexible deployment. The study of latency has been one of the research directions. The objective of [22] was to minimize the sum of the maximum delay among all the users in each slot. The trajectory of the UAV, the ratio of offloading task and the user scheduling variables were jointly optimized. In addition, energy efficient schemes have attracted wide attention due to the battery technology limitation of the mobile devices and the UAVs. In [23], the energy consumption of the mobile users was minimized considering two types of access schemes, namely, orthogonal and non-orthogonal access. The bits allocation and the trajectory of the UAV were jointly optimized. However, the energy efficient algorithms of the UAVs, which significantly influence the performance of the UAV-enabled MEC system, were not considered. The authors in [24] studied the minimization problem of the weighted sum energy consumption of the UAV and users. The computation resources scheduling, the bandwidth allocation and the trajectory of the UAV were optimized in the minimization problem. In [25], a UAV-enabled wireless powered MEC server was studied to minimize the energy consumption consumed at the UAV. In [26], the weighted sum computation bits were maximized under both the partial and binary offloading mode in UAV-enabled wireless powered MEC system. However, the energy consumption constraint of the UAV was not taken into consideration in [24,25,26]. The energy consumption of the UAV impacts the duration of flight and then impacts the performance of the UAV-enabled MEC system. Hence, it is necessary to design an energy-efficient UAV-enabled MEC systems considering both the energy consumption of the UAV and mobile devices.

### 1.2. Contribution and Organization

Different from the aforementioned study, this paper proposes an energy-efficient UAV-enabled MEC system with the objective of minimizing the total energy consumption of the system. Moreover, the constraints of the UAV energy budget, the number of each task’s bits, the causality of the data, and the velocity of the UAV are taken into consideration. The bits allocation and the trajectory of the UAV are jointly optimized under the constraints. Besides, an alternating algorithm is designed to solve the non-convex optimization problem. The main contributions are described as follows:An energy-efficient scheme subjecting to the constraints of UAV’s energy budget, the number of each task’s bits, the causality of the data, and the velocity limitation of the UAV is proposed. The aim of the problem is to minimize the total energy-consumption of the UAV-enabled MEC system, which includes the energy consumption of the ground users and the UAV. The bits allocation and the trajectory of the UAV are cooperatively optimized.A two-stage alternating algorithm is presented to solve the optimization problem. The formulated optimization problem is non-convex due to the non-convex objective function and non-convex constraints. The dual variables also enhance the difficulty of the problem. To solve the optimization problem, an alternating algorithm is proposed. The subproblems are solved by Lagrange duality method and CVX solver respectively.The simulation results are shown to indicate the superiority of the proposed scheme compared with other benchmark schemes. In contrast to the fixed trajectory of the UAV, the energy consumption of the proposed scheme is greatly decreased. Besides, the performance of the proposed scheme is also demonstrated under different settings.

The remainder of this paper is organized as follows. Section 2 sets up the system model and formulates the optimization problem. An alternating algorithm to solve the optimization problem is designed in Section 3. Then, the simulation results are shown in Section 4. Finally, the paper is concluded in Section 5.

## 2. System Model and Problem Formulation

The offloading modes of MEC include binary offloading and partial offloading [27]. In the first mode, the computation task cannot be partitioned, which can only be computed locally or offloaded wholely. In the last mode, a part of the computation task can be computed locally and the other part offloaded. In this paper, we only consider the binary offloading mode to relieve the computation load of the users. Considering the superiority of rotary-wing UAV’s agility in turning, the rotary-wing UAV is assumed to provide offloading services in this paper. The UAV-enabled MEC system is depicted in Figure 1, consisting of one UAV-enabled MEC server and *K* users denoted as K≜{1,2,3,⋯,K}. Moreover, taking the energy constraints of the UAV and the short task deadline of ground users into consideration, the area where the UAV-enabled MEC system can provide offloading services is not large. Thus, it is assumed that the ground users are not distributed over a large area and the ground users are always in the communication range of the UAV. The scenario that the ground users are distributed over a large scale will be further investigated in our future work.

### 2.1. System Model

The notation of user *k*’s task is denoted as $A_{k}\left(I_{k},\tau_{k},C_{k},O_{k}\right)$. $I_{k}$ means the task input-data size of user *k* in bits. $\tau_{k}$ means the task deadline of user *k*. $C_{k}$ represents the computation/intensity of user *k*, i.e., the number of CPU cycles needed to compute one input bit for user *k*. $O_{k}$ is the ratio of the number of output information bits to the number of input information bits for user *k*. The minimal task deadline of all the *K* users is considered as the time duration *T* of the UAV-enabled MEC system to finish the offloading tasks. The *K* users offload their computation tasks using a time division multiple access (TDMA) mode. The time duration *T* is divided into *N* slots, whose duration is $\Delta =\frac{T}{N}$. Each  slot Δ is divided into *K* sub-slots, whose duration is $\delta =\frac{\Delta }{K}=\frac{T}{NK}$. The slot and sub-slot are depicted as Figure 2. To reduce the additional noise from other users when user *k* is transmitting the data to the UAV and receiving the data from the UAV, user *k* only uploads its input information and the UAV only downloads the output results for user *k* in the *k*th sub-slot in each slot. The symbols used in the system model are listed in Table 1.

In this paper, a three-dimensional Euclidean coordinate is adopted without the loss of generality, which is measured in meters. The mobile users are located on the ground, whose coordinate is expressed as $q_{k}=\left(x_{k},y_{k},0\right)$ for k∈K. A UAV mounted with MEC server provides the computation services for the mobile users in the altitude *h*. The position of the UAV at the end of *n*th slot is expressed as $q_{u}\left[n\right]=\left(x_{u}\left[n\right],y_{u}\left[n\right],h\right)$, which is considered as the position of the UAV in *n*th slot. We assume that the communication link between the UAV and the ground user is dominated by the line of sight (LoS) channel. The UAV-ground channel is more likely to have the LoS link compared to the terrestrial ground-ground channels. Besides, the Doppler effect due to the UAV mobility is assumed to be compensated [17,28]. Hence, the channel gain from the ground user *k* to the UAV in the *n*th slot follows the free-space loss model, which is given by [29,30]
(1)$$g_{k}\left[n\right]=\frac{g_{0}}{h^{2}+{∥q_{u}\left[n\right]-q_{k}∥}^{2}},$$
where $g_{0}$ is the channel power gain at reference distance 1 m. From (Equation 1), it can be observed that optimizing the 2D position of the UAV ($x_{k},y_{k}$) has the same effect of optimizing the altitude of the UAV. Therefore, for ease of explanation, we assume that the UAV fly at a fixed altitude *H* and ingore the slight altitude changes.

The energy consumption of the UAV-enabled MEC system is consisted of two parts: the energy consumption of the UAV and the energy consumption of the ground users. Hence, the total energy consumption of the system *E* is the sum of the energy consumption of the UAV and the energy consumption of the ground users, which is given as:
(2)$$E=E_{G}+E_{U}.$$

#### 2.1.1. Energy Consumption Model of Ground Users

In the binary mode of the UAV-enabled MEC system, all the *K* users upload their computation tasks to the UAV and the ground users have no local computing tasks. Consequently, the energy consumption of the ground users $E_{G}$ only depends on the communication energy consumption for uploading data of the ground users $E_{G}^{U}$ as given [30,31]:(3)$$E_{G}=E_{G}^{U}=\sum_{n=1}^{N}\sum_{k=1}^{K}\frac{2^{\frac{I_{k}^{u}\left[n\right]}{B\delta }}-1}{g_{k}\left[n\right]}\sigma^{2}\delta ,$$
where *B* and $\sigma^{2}$ denote the communication bandwidth and the noise power at the receiver respectively; $I_{k}^{u}\left[n\right]$ represents the number of uploading input bits of the user *k* in the *n*th slot.

#### 2.1.2. Energy Consumption Model of the UAV

The energy consumption of the UAV consists of three parts: the energy consumption of computation $E_{U}^{C}$, the energy consumption of flying $E_{U}^{F}$ and the energy consumption of downloading the output results to the ground users $E_{U}^{D}$, which is shown as
(4)$$E_{U}=E_{U}^{C}+E_{U}^{F}+E_{U}^{D},$$

After the ground users uploading the input data to the UAV, the UAV will execute the computation immediately. The frequency of the UAV’s CPU in the *n*th slot for computing the tasks of user *k* depends on the number of computing bits for user *k* in the *n*th slot, which is expressed as
(5)$$f_{U,k}\left[n\right]=\frac{I_{k}^{c}\left[n\right]C_{k}}{\Delta },$$
where the $I_{k}^{c}\left[n\right]$ is the number of the computing bits at the UAV in the *n*th slot for ground user *k*. The computing energy consumption of the UAV depends on the frequency of the CPU. Thus, the computation energy consumption of the UAV for ground user *k* in the *n*th slot is shown as
(6)$$E_{U,k}^{C}\left[n\right]=\Delta \gamma_{u}{\left(f_{U,k}\left[n\right]\right)}^{3}=\gamma_{u}\frac{{\left(I_{k}^{c}\left[n\right]\right)}^{3}C_{k}^{3}}{\Delta^{2}},$$
where $\gamma_{u}$ is the effective switched capacitance of the UAV’s CPU [23,25,26,32,33]. The total energy consumption of computation in all slots of the UAV is calculated as
(7)$$E_{U}^{C}=\sum_{n=1}^{N}\sum_{k=1}^{K}\gamma_{u}\frac{{\left(I_{k}^{c}\left[n\right]\right)}^{3}C_{k}^{3}}{\Delta^{2}}.$$

The flying energy consumption of the UAV in slot *n* depends on the velocity v[n] and the weight of the UAV *M*. The total energy consumption for propulsion of the UAV in all slots is given as
(8)$$E_{U}^{F}=\sum_{n=1}^{N}\kappa {∥v\left[n\right]∥}^{2},$$
where κ=0.5MΔ and *M* is the weight of the UAV [34,35,36,37]. The velocity of the UAV in *n*th slot can be calculated as the distance difference in *n*th slot divided by the time duration of one slot Δ. And the distance difference of the UAV in *n*th slot is related to the position of the UAV in *n*th slot $q_{u}\left[n\right]$ and in (n−1)th slot $q_{u}\left[n-1\right]$. Thus, the velocity of the UAV in *n*th slot v[n] is expressed as
(9)$$v\left[n\right]=\frac{q_{u}\left[n\right]-q_{u}\left[n-1\right]}{\Delta },\;n=1,\;2,\cdots ,\;N.$$
And the value of the UAV’s velocity in *n*th slot should be lower than the maximal velocity of the UAV as expressed as
(10)$$∥v\left[n\right]∥=\frac{∥q_{u}\left[n\right]-q_{u}\left[n-1\right]∥}{\Delta }\leq V_{max},\;n=1,\;2,\cdots ,\;N.$$
where $V_{max}$ denotes the maximal velocity of the UAV. The energy consumption of downloading the output results from the UAV to the ground users is related to the number of downloading bits and the channel gain, which is given as
(11)$$E_{U}^{D}=\sum_{n=1}^{N}\sum_{k=1}^{K}\frac{2^{\frac{I_{k}^{d}\left[n\right]}{B\delta }}-1}{g_{k}\left[n\right]}\sigma^{2}\delta ,$$
where $I_{k}^{d}\left[n\right]$ denotes the number of the downloading bits from the UAV to the ground user *k* in the *n*th slot.

### 2.2. Problem Formulation

The objective of the formulated problem is to minimize the total energy consumption of the system, by joint optimizing the trajectory of the UAV $q_{u}\left[n\right]$ and the bits allocation $I_{k}^{u}\left[n\right]$, $I_{k}^{c}\left[n\right]$, $I_{k}^{d}\left[n\right]$. In this paper, the initial position of the UAV is preset as the position of $q_{1}$, and the final point of the UAV is the position of $q_{K}$, which are expressed as
(12)$$q_{u}\left[1\right]=q_{1},q_{u}\left[n\right]=q_{k}.$$

All users offload their computation tasks to the UAV and the computing results should be distributed to ground users in the duration of *N* slots. There are three steps for user *k* to offload the task to the UAV: the ground user *k* uploads the task to the UAV, the UAV processes data for user *k*, and the UAV downloads the computation results to the ground user *k*, as shown in Figure 3. These  three steps are conducted in time sequence. Consequently, the ground users do not upload the input information in the last two slots. The UAV does not execute the computation tasks in 1st slot and *N*th slot. Moreover, the UAV does not download the computation results to the ground users in 1st slot and 2nd slot. Thus, considering the causality of the data and the chronological order, the bit allocations follow the constraints:(13)$$\sum_{i=1}^{n-1}I_{k}^{u}\left[i\right]\geq \sum_{i=2}^{n}I_{k}^{c}\left[i\right],n=2,\;\cdots ,\;N-1,$$
(14)$$O_{k}\sum_{i=2}^{n}I_{k}^{c}\left[i\right]\geq \sum_{i=3}^{n+1}I_{k}^{d}\left[i\right],n=2,\;\cdots ,\;N-1,$$
(15)$$I_{k}^{u}\left[N-1\right]=I_{k}^{u}\left[N\right]=0,$$
(16)$$I_{k}^{c}\left[1\right]=I_{k}^{c}\left[N\right]=0,$$
(17)$$I_{k}^{d}\left[1\right]=I_{k}^{d}\left[2\right]=0.$$

Moreover, the total number of the uploading bits of the ground user *k*, computing bits and downloading bits of the UAV for the user *k* are expressed as follows:(18)$$\sum_{n=1}^{N-2}I_{k}^{u}\left[n\right]=I_{k},$$
(19)$$\sum_{n=2}^{N-1}I_{k}^{c}\left[n\right]=I_{k},$$
(20)$$\sum_{n=3}^{N}I_{k}^{d}\left[n\right]=O_{k}I_{k}.$$

The objective of the energy-efficient problem is to minimize the total energy consumption of the UAV-enable MEC system. The bit allocations $I_{k}^{u}\left[n\right]$, $I_{k}^{c}\left[n\right]$, $I_{k}^{d}\left[n\right]$ and the trajectory of the UAV $q_{u}\left[n\right]$ are jointly optimized, under the constraints of data causality, the velocity and the position of the UAV. The corresponding energy consumption minimization problem P1 is formulated as
(21a)$$\begin{array}{cc} \; & P1:\;\min_{I_{k}^{u}\left[n\right],I_{k}^{c}\left[n\right],I_{k}^{d}\left[n\right],q_{u}\left[n\right]}E \end{array}$$
(21b)$$\begin{array}{cc} \; & s.t.\;E_{U}\leq \varepsilon , \end{array}$$
(21c)$$\begin{array}{cc} \; & \sum_{n=1}^{N-2}I_{k}^{u}\left[n\right]=I_{k}, \end{array}$$
(21d)$$\begin{array}{cc} \; & \sum_{n=2}^{N-1}I_{k}^{c}\left[n\right]=I_{k}, \end{array}$$
(21e)$$\begin{array}{cc} \; & \sum_{n=3}^{N}I_{k}^{d}\left[n\right]=O_{k}I_{k}, \end{array}$$
(21f)$$\begin{array}{cc} \; & \sum_{i=1}^{n-1}I_{k}^{u}\left[i\right]\geq \sum_{i=2}^{n}I_{k}^{c}\left[i\right],\;n=2,\;\cdots ,\;N-1, \end{array}$$
(21g)$$\begin{array}{cc} \; & O_{k}\sum_{i=2}^{n}I_{k}^{c}\left[i\right]\geq \sum_{i=3}^{n+1}I_{k}^{d}\left[i\right],\;n=2,\;\cdots ,\;N-1, \end{array}$$
(21h)$$\begin{array}{cc} \; & I_{k}^{u}\left[N-1\right]=I_{k}^{u}\left[N\right]=0, \end{array}$$
(21i)$$\begin{array}{cc} \; & I_{k}^{c}\left[1\right]=I_{k}^{c}\left[N\right]=0, \end{array}$$
(21j)$$\begin{array}{cc} \; & I_{k}^{d}\left[1\right]=I_{k}^{d}\left[2\right]=0, \end{array}$$
(21k)$$\begin{array}{cc} \; & I_{k}^{u}\left[n\right],I_{k}^{c}\left[n\right],I_{k}^{d}\left[n\right]\geq 0,\;\mathrm{for}\;k\in K\;\mathrm{and}\;n\in N. \end{array}$$
(21l)$$\begin{array}{cc} \; & q_{u}\left[0\right]=q_{1},q_{u}\left[N\right]=q_{K}, \end{array}$$
(21m)$$\begin{array}{cc} \; & \frac{∥q_{u}\left[n\right]-q_{u}\left[n-1\right]∥}{\Delta }\leq V_{max},\;\mathrm{for}\;n\in N, \end{array}$$
where ε denotes the energy budget of the UAV, and N≜{1,⋯,N}. (21b) indicates that the energy consumption of the UAV, including the computing energy consumption, communication energy consumption and flying energy consumption, is lower than a certain energy budget. (21c), (21d) and (21e) represent that all the ground users offload all the computation tasks to the UAV in the binary mode. (21f)–(21k) ensure the data causality and the non-negative condition. (21l) guarantees that the UAV initiates the flying trajectory at the position of $q_{1}$ and stops at the position of $q_{K}$. (21m) ensures that the UAV flies under the constraint of the maximal velocity.

## 3. Algorithm Design

The problem of P1 is a non-convex problem due to the non-convex objective function ([Disp-formula FD21a-sensors-19-04521]), non-convex constraint (21b) and the coupling characteristics of the optimization variables. Thus, in this paper, we bring forward a two-stage alternating algorithm to solve the non-convex problem. The details of the two-stage alternating algorithm are shown in the following part.

### 3.1. Bits Allocation under Given Trajectory

It can be seen that with if the trajectory of the UAV is fixed, the problem of P1 is convex with regard to the variables $I_{k}^{u}\left[n\right],I_{k}^{c}\left[n\right],I_{k}^{d}\left[n\right]$. Given the trajectory of the UAV, the problem P1 can be transformed into:
(22a)$$\begin{array}{cc} \; & P2:\;\min_{I_{k}^{u}\left[n\right],I_{k}^{c}\left[n\right],I_{k}^{d}\left[n\right]}E_{U}+E_{G} \end{array}$$
(22b)$$\begin{array}{cc} \; & s.t.\;(21b)-(21k). \end{array}$$

Because P2 is convex, Lagrange duality method can be applied [38]. The optimal solution of P2 is given in the following theorem. $I_{k}^{u*}\left[n\right]$, $I_{k}^{c*}\left[n\right]$ and $I_{k}^{d*}\left[n\right]$ represent the optimal number of uploading bits of user *k* in *n*th slot, the optimal number of computing bits of the UAV in *n*th slot for user *k* and the optimal number of downloading bits from the UAV to the user *k* in *n*th slot respectively under the given trajectory of the UAV.

**Theorem** **1.**
*For a given trajectory of the UAV, the optimal bits allocation $I_{k}^{u}\left[n\right]$, $I_{k}^{c}\left[n\right]$, $I_{k}^{d}\left[n\right]$ can be expressed  as:*
(23a)$$\begin{array}{cc} \; & I_{k}^{u*}\left[n\right]=\left\{\begin{array}{c} B\delta \log_{2}\frac{\left(\mu_{k}+\sum_{i=n+1}^{N-1}\beta_{k,n}\right)g_{k}\left[n\right]B}{\sigma^{2}\ln2},\;n=1,\;\cdots \;,N-2,\\ 0,\;n=N-1\;\mathrm{or}\;N, \end{array}\right. \end{array}$$
(23b)$$\begin{array}{cc} \; & I_{k}^{c*}\left[n\right]=\left\{\begin{array}{c} \frac{\Delta }{C_{k}}\sqrt{\frac{\nu_{k}+O_{k}\sum_{i=n}^{N-1}\eta_{k,n}-\sum_{i=n}^{N-1}\beta_{k,n}}{3\left(\lambda +1\right)\gamma_{u}C_{k}}},\;n=2,\;\cdots ,\;N-1,\\ 0,\;n=1\;\mathrm{or}\;N, \end{array}\right. \end{array}$$
(23c)$$\begin{array}{cc} \; & I_{k}^{d*}\left[n\right]=\left\{\begin{array}{c} B\delta \log_{2}\frac{\left(\rho_{k}-\sum_{i=n-1}^{N-1}\eta_{k,n}\right)g_{k}\left[n\right]B}{\left(\lambda +1\right)\ln2\sigma^{2}},\;n=3,\;\cdots ,\;N,\\ 0,\;n=1\;\mathrm{or}\;2. \end{array}\right. \end{array}$$
*where λ, $\mu_{k}$, $\nu_{k}$, $\rho_{k}$, $\beta_{k,n}$, $\eta_{k,n}$ denote the dual variable related to the constraints *(21b)–(21g)* respectively.*


**Proof.** See Appendix A. ☐

The dual variables $\lambda ,\mu_{k},\nu_{k},\rho_{k},\beta_{k,n},\eta_{k,n}$ in ([Disp-formula FD23a-sensors-19-04521])–(23c) are obtained with the subgradient algorithm [39] as shown in Lemma 1.

**Lemma** **1.**
*At the (j+1)th iteration in the subgradient method, $\lambda_{j+1}$, $\mu_{k,j+1}$, $\nu_{k,j+1}$, $\rho_{k,j+1}$, $\beta_{k,n,j+1}$, $\eta_{k,n,j+1}$ are obtained from*
(24a)$$\begin{array}{cc} \; & \lambda_{k,j+1}=\lambda_{k,j}-\alpha_{j}^{\left(\lambda \right)}g_{j}^{\left(\lambda \right)}, \end{array}$$
(24b)$$\begin{array}{cc} \; & \mu_{k,j+1}=\mu_{k,j}-\alpha_{j}^{\left(\mu \right)}g_{j}^{\left(\mu \right)}, \end{array}$$
(24c)$$\begin{array}{cc} \; & \nu_{k,j+1}=\nu_{k,j}-\alpha_{j}^{\left(\nu \right)}g_{j}^{\left(\nu \right)}, \end{array}$$
(24d)$$\begin{array}{cc} \; & \rho_{k,j+1}=\rho_{k,j}-\alpha_{j}^{\left(\rho \right)}g_{j}^{\left(\rho \right)}, \end{array}$$
(24e)$$\begin{array}{cc} \; & \beta_{k,n,j+1}=\beta_{k,n,j}-\alpha_{j}^{\left(\beta \right)}g_{n,j}^{\left(\beta \right)}, \end{array}$$
(24f)$$\begin{array}{cc} \; & \eta_{k,n,j+1}=\eta_{k,n,j}-\alpha_{j}^{\left(\eta \right)}g_{n,j}^{\left(\eta \right)}, \end{array}$$
*where $\alpha_{j}^{\left(\lambda \right)}$, $\alpha_{j}^{\left(\mu \right)}$, $\alpha_{j}^{\left(\nu \right)}$, $\alpha_{j}^{\left(\rho \right)}$, $\alpha_{j}^{\left(\beta \right)}$ and $\alpha_{j}^{\left(\eta \right)}$ denote the jth step size respectively. The corresponding subgradients $g_{j}^{\left(\lambda \right)}$, $g_{j}^{\left(\mu \right)}$, $g_{j}^{\left(\nu \right)}$, $g_{j}^{\left(\rho \right)}$, $g_{n,j}^{\left(\beta \right)}$ and $g_{n,j}^{\left(\eta \right)}$ are given as*
(25a)$$\begin{array}{c} \begin{array}{cc} \; & g_{j}^{\left(\lambda \right)}=\sum_{n=1}^{N}\sum_{k=1}^{K}\gamma_{u}C_{u}^{4}\frac{{\left(I_{k,j}^{c*}\left[n\right]\right)}^{3}}{\Delta^{3}}+\sum_{n=1}^{N}\sum_{k=1}^{K}\frac{2^{\frac{I_{k,j}^{d}*\left[n\right]}{B\Delta }}-1}{g_{k}\left[n\right]}\sigma^{2}\Delta +\\ \; & \sum_{n=1}^{N}{∥v\left[n\right]∥}^{2}-\varepsilon , \end{array} \end{array}$$
(25b)$$\begin{array}{cc} \; & g_{j}^{\left(\mu \right)}=I_{k}-\sum_{n=1}^{N-1}I_{k,j}^{u*}\left[n\right], \end{array}$$
(25c)$$\begin{array}{cc} \; & g_{j}^{\left(\nu \right)}=I_{k}-\sum_{n=2}^{N-1}I_{k,j}^{c*}\left[n\right], \end{array}$$
(25d)$$\begin{array}{cc} \; & g_{j}^{\left(\rho \right)}=O_{k}I_{k}-\sum_{n=3}^{N}I_{k,j}^{d*}\left[n\right], \end{array}$$
(25e)$$\begin{array}{cc} \; & g_{n,j}^{\left(\beta \right)}=\sum_{i=2}^{n}I_{k,j}^{c*}\left[i\right]-\sum_{i=1}^{n-1}I_{k,j}^{u*}\left[i\right], \end{array}$$
(25f)$$\begin{array}{cc} \; & g_{n,j}^{\left(\eta \right)}=\sum_{i=2}^{n}I_{k,j}^{d*}-O_{k}\sum_{i=2}^{n-1}I_{k,j}^{c*}\left[i\right], \end{array}$$
*where $I_{k,j}^{u*}$, $I_{k,j}^{c*}$ and $I_{k,j}^{d*}$ denote the optimal bits allocation at the jth iteration which are obtained from *([Disp-formula FD23a-sensors-19-04521])–(23c)*.*


### 3.2. Trajectory Design under Given Bits Allocation

When the bits allocation of each slot $I_{k}^{u}\left[n\right]$, $I_{k}^{c}\left[n\right]$ and $I_{k}^{d}\left[n\right]$ are given, the problem P1 is transformed  into
(26a)$$\begin{array}{cc} \; & P3:\;\min_{v[n]}E_{U}+E_{G}, \end{array}$$
(26b)$$\begin{array}{cc} \; & s.t.\;(21b),\;(21l)\;\mathrm{and}\;(21m). \end{array}$$

It can be seen that the objective function and the constraints of **P3** are convex, and the duality of the variables is weak. Then, **P3** can be solved by the CVX solver [38,40].

A two-stage alternating algorithm is proposed based on the solution of **P2** and **P3**. The details of the alternating algorithm are shown in Algorithm 1, where $E_{j}^{i}$ and $E^{i}$ denote the total energy consumption of the UAV-enabled MEC system in each iteration as calculated in (Equation 2).

**Algorithm 1.** The alternating algorithm for **P1**.**Input:***K*, *N*, *B*, Δ, δ$O_{k}$, $C_{k}$, κ, $\gamma_{u}$, $\sigma^{2}$, $g_{0}$, $V_{max}$, ε$q_{k}$, and tolerant thresholds ξ and $\xi_{1}$;

1: 2:**Initialize:** iterative number i=1 and $q_{u}^{1}\left[n\right]$;3: 4:
**repeat**
5: 6:    **Initialize:** iterative number j=1 and $\lambda_{1}$, $\mu_{k,1}$, $\nu_{k,1}$, $\rho_{k,1}$, $\beta_{k,n,1}$, $\eta_{k,n,1}$;7: 8:    **repeat**9: 10:        Calculate the $I_{k,j}^{u,i*}\left[n\right]$, $I_{k,j}^{c,i*}\left[n\right]$ and $I_{k,j}^{d,i*}\left[n\right]$ by Theorem 1 under the given trajectory $q_{u}^{i}\left[n\right]$;11: 12:        Update the iterative number j=j+1;13: 14:        Update $\lambda_{j}$, $\mu_{k,j}$, $\nu_{k,j}$, $\rho_{k,j}$, $\beta_{k,n,j}$, $\eta_{k,n,j}$ through Lemma 1;15: 16:    **until**
$E_{j}^{i}-E_{j-1}^{i}\leq \xi_{1}$(j>1);17: 18:    Let $I_{k}^{u,i*}=I_{k,j}^{u,i*}$, $I_{k}^{c,i*}=I_{k,j}^{c,i*}$ and $I_{k}^{d,i*}=I_{k,j}^{d,i*}$;19: 20:    Solve **P3** with the CVX solver and obtain $q_{u}^{i*}\left[n\right]$ under the given bits allocation $I_{k}^{u,i*}$, $I_{k}^{c,i*}$ and $I_{k}^{d,i*}$;21: 22:    Let $q_{u}^{i}\left[n\right]$=$q_{u}^{i*}\left[n\right]$;23: 24:    Update the iterative number i=i+1;25: 26:**until** $E^{i}-E^{i-1}\leq \xi$(i>1);27: 28:Let $E=E^{i}$, $I_{k}^{u}\left[n\right]=I_{k}^{u,i*}$, $I_{k}^{c}\left[n\right]=I_{k}^{c,i*}$, $I_{k}^{d}\left[n\right]=I_{k}^{d,i*}$ and $q_{u}\left[n\right]=q_{u}^{i}\left[n\right]$;29: 
**Output:***E*, $I_{k}^{u}\left[n\right]$, $I_{k}^{c}\left[n\right]$, $I_{k}^{d}\left[n\right]$ and $q_{u}\left[n\right]$.
30:

### 3.3. Complexity Analysis

The Complexity of Algorithm 1 comes from three parts. The first part is from the computation of $I_{k}^{u*}\left[n\right]$, $I_{k}^{c*}\left[n\right]$ and $I_{k}^{d*}\left[n\right]$ in Theorem 1. The second part comes from the subgradient method for computing the dual variables. The third part is from the application of CVX to solve the trajectory of the UAV. Let *L* denote the number of interations of outer loop. According to the work in [38,41], the complexity of Algorithm 1 is O [$L(KN+\frac{1}{\xi_{1}^{2}}+N^{3})$] Generally, small values of *K* and *N* are enough for good performance. Thus the proposed algorithm is feasible in practice.

## 4. Simulation Results

In this section, we evaluate the performance of the proposed algorithm by simulation. The parameters of the simulation are shown in Table 2 [22,23,24,25,26].

We consider five trajectories of the UAV to compare the energy consumption of the UAV-enabled MEC system. The size of input tasks of each user are assumed to be $I_{1}=4\times 10^{7}$ bits, $I_{2}=5\times 10^{7}$ bits, $I_{3}=6\times 10^{7}$ bits, $I_{4}=7\times 10^{7}$ bits and $I_{5}=8\times 10^{7}$ bits respectively. All tasks should be processed within T=5 s. The duration of each slot and sub-slot are set as $5\times 10^{-2}$ s and $10^{-2}$ s separately. The positions of each user are $p_{1}=\left(0,0,0\right)$, $p_{2}=\left(16,0,0\right)$, $p_{3}=\left(8,8,0\right)$, $p_{4}=\left(16,16,0\right)$ and $p_{5}=\left(0,16,0\right)$. The starting position of the UAV is (0,0,H) and the final position is (0,16,H). The simulation compares five trajectories: the trajectory using the proposed algorithm, the square trajectory whose length is 16m, the semicircle whose radius is 8m, the line trajectory from (0,0,H) to (0,16,H) and the fixed point in (8,8,H), as shown in Figure 4.

The total energy consumption of four pre-defined trajectories and the proposed algorithm are compared under the same parameter settings. In four pre-defined trajectories, the bits allocation is optimized under given trajectory. The total energy consumption of the UAV-enabled system under five trajectories are $1.8468\times 10^{4}$ J, $1.9812\times 10^{4}$ J, $1.8735\times 10^{4}$ J, $1.9343\times 10^{4}$ J and $1.8665\times 10^{4}$ J respectively, as given in Figure 5. The trajectory of the proposed scheme tends to user 2, user 3 and user 4 to receive the input data and transmit output results. Compared with the pre-defined trajectories, the trajectory of the proposed scheme is optimized to minimize the total energy consumption. In comparison with four pre-defined trajectories, the proposed scheme performs the lowest system energy consumption as shown in Figure 5. It is not necessary for the UAV to go through every ground user to provide MEC services. Consequently, the energy consumption of the square trajectory exceeds the proposed scheme. In the semicircle trajectory, the UAV flies beyond the proposed scheme, causing the augment of the energy consumption of mobility. Although the length of the line trajectory and fixed point decrease compared with the length of the proposed scheme, the energy consumption of communication leads to the increase of the total energy consumption of the system.

Figure 6 shows the optimized bits allocation of each user in each slot for the trajectory using the proposed algorithm of Figure 4. It can be seen from Figure 6 that the tendency of the number of uploading bits, computing bits and downloading bits are similar. The number of uploading bits is relatively high in the beginning slots and decreases over time. Therefore, the UAV can receive all the tasks and provide the input data for the MEC server timely. Then, the UAV deals with the input data and the number of computing bits maintains a steady value. When the computing results output from the MEC server, the UAV needs to transmit the results to the ground users with time constraints. Therefore, the number of the downloading bits increases over time. The number of downloading bits is relatively low compared with the number of uploading bits because of the output/input ratio $O_{k}$.

The trajectory of the UAV varies under different time duration constraints for the proposed scheme. Figure 7 shows the trajectories of the UAV under different time constraints 4 s, 4.5 s and 5 s, assuming that $I_{1}=4\times 10^{7}$ bits, $I_{2}=5\times 10^{7}$ bits, $I_{3}=6\times 10^{7}$ bits, $I_{4}=7\times 10^{7}$ bits and $I_{5}=8\times 10^{7}$ bits. When T=5 s, the trajectory of the UAV tends to user 2, user 3 and user 4 to receive the input data and download the computing results. As the time constraint decreases, the trajectory of the UAV shrinks to user 1 and user 2, because the velocity of the UAV is limited and the UAV should complete the offloading tasks under the finite time duration constraint. Figure 8 shows the velocities of the UAV for the trajectories shown in Figure 7 under different time constraints. The UAV flies fast at the start to fly forward to the ground users to receive the uploading data. Then, the UAV reduces the speed over time to ensure that all ground users can upload the data and the UAV has time to process the data. Finally, the UAV speeds up in the end to finish the offloading tasks and to reach the destination timely. Moreover, as the time constraint decreases, the velocity of the UAV increases to complete the tasks under the limited task deadline.

In the two-stage alternating algorithm, the subproblem at a certain iteration is solved given the optimized variables derived from last iteration or last stage. So the alternating algorithm proposed in this paper should perform convergence. Figure 9 shows the convergence of the total energy consumption of the UAV-enabled MEC system under different time constraints when $I_{1}=4\times 10^{7}$ bits, $I_{2}=5\times 10^{7}$ bits, $I_{3}=6\times 10^{7}$ bits, $I_{4}=7\times 10^{7}$ bits and $I_{5}=8\times 10^{7}$ bits. It can be seen from Figure 9 that the total energy consumption of the UAV-enabled MEC system converges to a certain value using the proposed algorithm. Only a few iteration numbers are needed to converge by applying the alternating algorithm. Besides, the total energy consumption rises as the time constraint decreases, because the increase of the UAV’s velocity leads to the growth of the total energy consumption under the limited task deadline.

The superiority of the proposed algorithm is also demonstrated. In Figure 10, the proposed two-stage alternating scheme and the successive convex approximation (SCA) scheme are compared. The SCA algorithm is used to tackle the non-convex problems by applying the convex approximation of the non-convex objective and constraints [23,42,43]. For the generality, the users are assumed to be randomly distributed. Under the same distribution of the ground users, the optimizing trajectories of the UAV using the proposed scheme and the SCA scheme are compared. Figure 11 shows the total energy consumption of the UAV-enabled MEC system using the proposed two-stage alternating algorithm and the SCA algorithm, assuming that T=5 s, $I_{1}=I_{3}=I_{5}=2\times 10^{7}$ bits and $I_{2}=I_{4}=8\times 10^{7}$ bits. It can be seen from Figure 11 that the proposed scheme outperforms the SCA scheme in terms of the total energy consumption of the UAV-enabled MEC system. The UAV under the SCA algorithm flies further than the proposed algorithm as shown in Figure 10. The longer distance leads to the larger energy consumption of the UAV and the UAV-enabled MEC system.

In Figure 11, the energy consumption of the system using the proposed algorithm, the energy consumption of the ground users using the proposed algorithm, the energy consumption of the system using the SCA algorithm and the energy consumption of the ground users by computing the tasks locally are shown respectively. The energy consumption is compared under different time constraints and the same user distribution as shown in Figure 10, assuming that T=5 s, $I_{1}=I_{3}=I_{5}=2\times 10^{7}$ bits and $I_{2}=I_{4}=8\times 10^{7}$ bits. Figure 11 demonstrates that the proposed algorithm performs lower energy consumption of the system than the SCA algorithm under different time constraints. Moreover, from Figure 11, we observe that offloading the computing tasks to the UAV-enabled MEC system alleviates the burden of the ground users greatly. We also note that as the time constraint becomes more stringent, the energy savings of the UAV-enabled system using the proposed scheme becomes more prominent compared with computing locally. It should be noted that if the time constraint is too short for the UAV to complete the offloading tasks under the velocity limit, the optimization problem becomes infeasible.

To show the comlexity of Algorithm 1, the run time of Algorithm 1 is given in Table 3. The run time is obtained by using a computer with 64-bit Intel(R) i5-4210 CPU, 8GB RAM. From Table 3, it can be seen that it is feasible to implement Algorithm 1. The complexity of Algorithm 1 is affected by the number of slots *N* and the number of ground users *K*. And the table shows that *N* has a larger influence on the complexity than *K*, as analyzed in Section 4. Besides, how to reduce the complexity of the algorithm, especially when *N* increases, will be further studied in our future work.

## 5. Conclusions

In this paper, a UAV-enabled MEC system is studied in which the UAV carries computing resources to provide computation offloading services for the ground users. The bits allocation of uploading data, computing data, downloading data and the trajectory of the UAV are jointly optimized with the goal of minimizing the total energy consumption of the UAV-enabled system, including the flying energy consumption and the transmission energy consumption. The energy consumption minimization problem is optimized under the constraints of the number of each task’s bits, the causality of the data and the velocity of the UAV. On account of the fact that the optimization problem is non-convex, an alternating algorithm is proposed in this paper to solve the formulated problem. The simulation results illustrate that the proposed scheme is superior to the benchmark schemes. Moreover, the simulation results also show the performance of the proposed scheme with different parameters. In the future work, we will consider the more complicated models for flying and communications, such as the fading factors and the mobility of ground users.

## Figures and Tables

**Figure 1 sensors-19-04521-f001:**
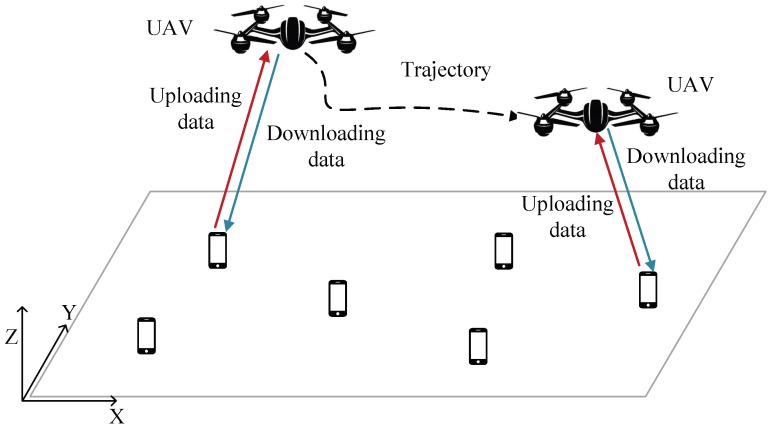
UAV-enabled MEC system.

**Figure 2 sensors-19-04521-f002:**
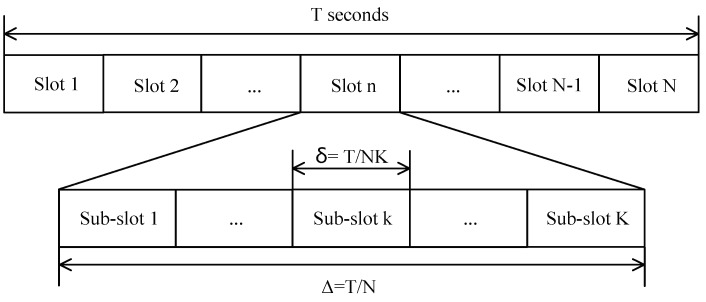
The slots and sub-slots in time division multiple access (TDMA).

**Figure 3 sensors-19-04521-f003:**
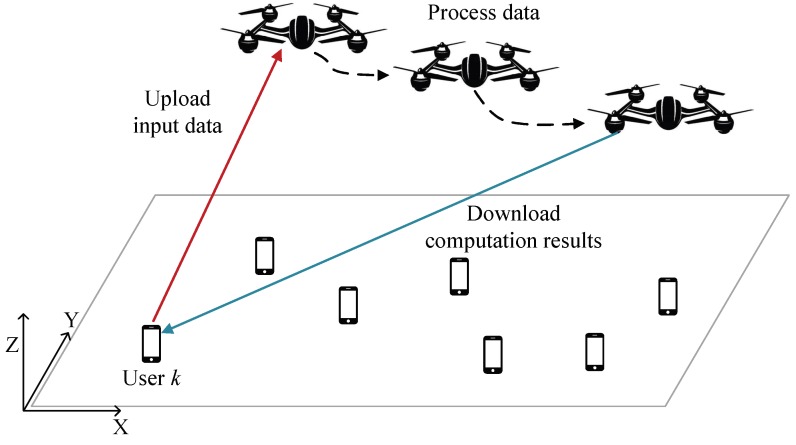
The slots and sub-slots in TDMA.

**Figure 4 sensors-19-04521-f004:**
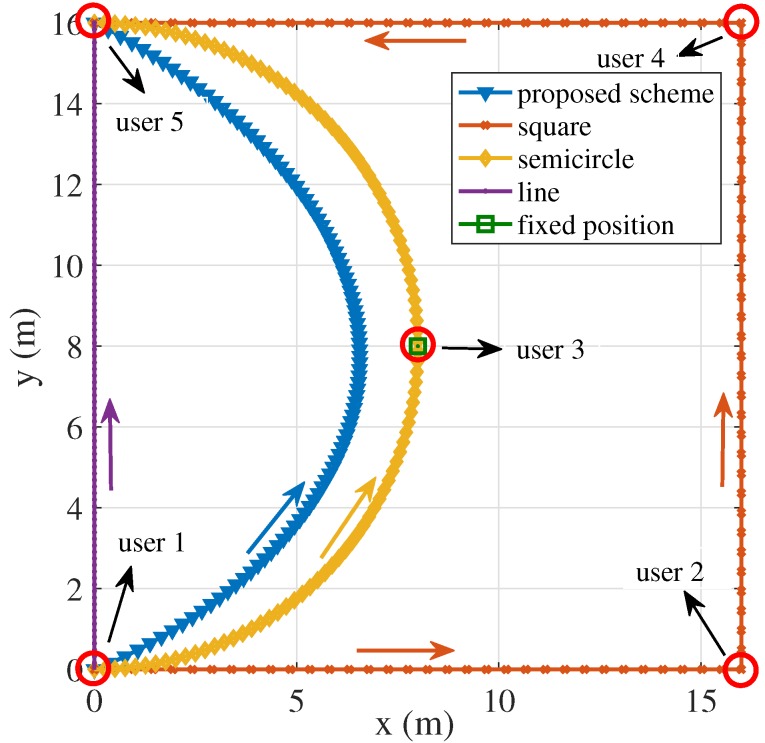
Different trajectory of the Unmanned Aerial Vehicle (UAV) under the time constraint T=5 s ($I_{1}=4\times 10^{7}$ bits, $I_{2}=5\times 10^{7}$ bits, $I_{3}=6\times 10^{7}$ bits, $I_{4}=7\times 10^{7}$ bits and $I_{5}=8\times 10^{7}$ bits).

**Figure 5 sensors-19-04521-f005:**
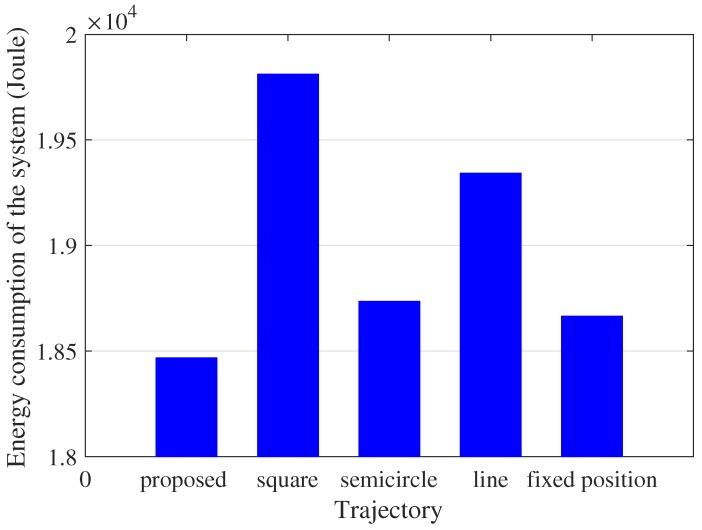
Energy consumption of the UAV-enabled mobile edge computing (MEC) system in different trajectory.

**Figure 6 sensors-19-04521-f006:**
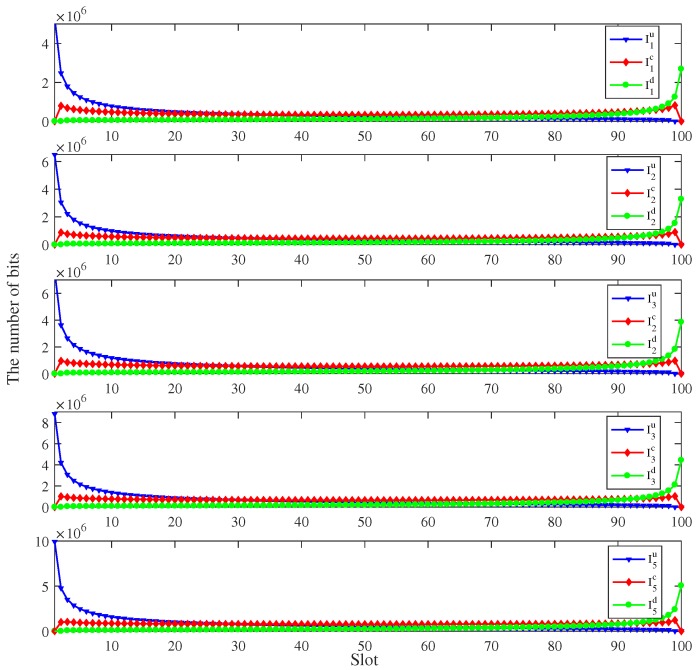
The bits allocation of all users under the time constraint T=5 s, $I_{1}=4\times 10^{7}$ bits, $I_{2}=5\times 10^{7}$ bits, $I_{3}=6\times 10^{7}$ bits, $I_{4}=7\times 10^{7}$ bits and $I_{5}=8\times 10^{7}$ bits.

**Figure 7 sensors-19-04521-f007:**
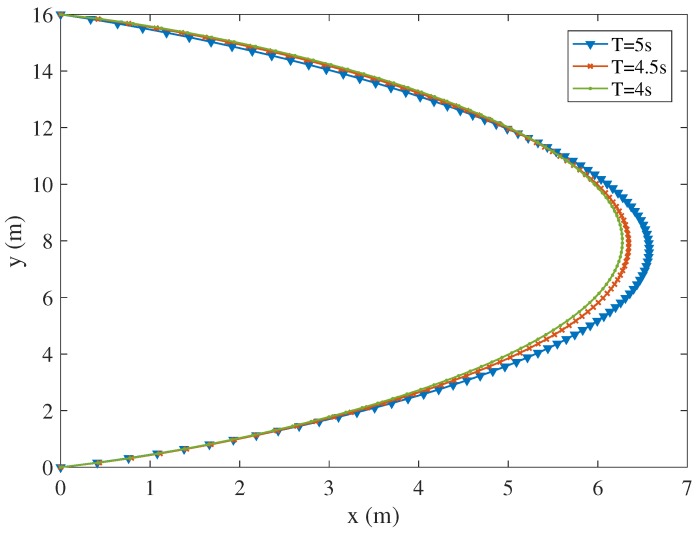
The trajectory of the UAV under different time constraint ($I_{1}=4\times 10^{7}$ bits, $I_{2}=5\times 10^{7}$ bits, $I_{3}=6\times 10^{7}$ bits, $I_{4}=7\times 10^{7}$ bits and $I_{5}=8\times 10^{7}$ bits).

**Figure 8 sensors-19-04521-f008:**
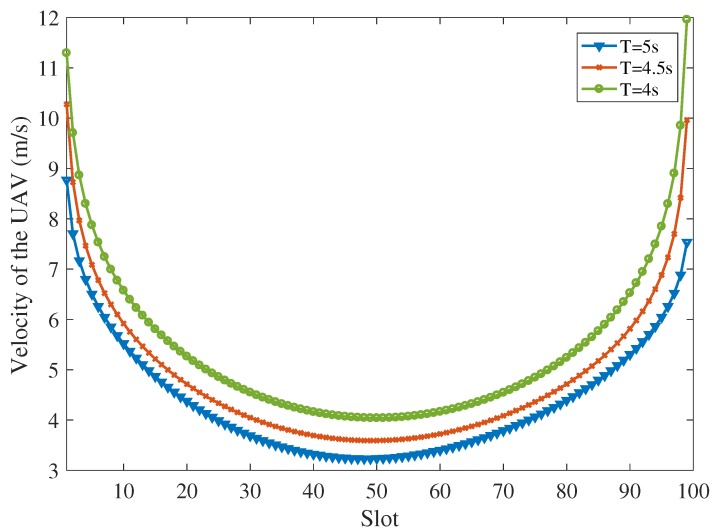
The velocity of the UAV ($I_{1}=4\times 10^{7}$ bits, $I_{2}=5\times 10^{7}$ bits, $I_{3}=6\times 10^{7}$ bits, $I_{4}=7\times 10^{7}$ bits and $I_{5}=8\times 10^{7}$ bits).

**Figure 9 sensors-19-04521-f009:**
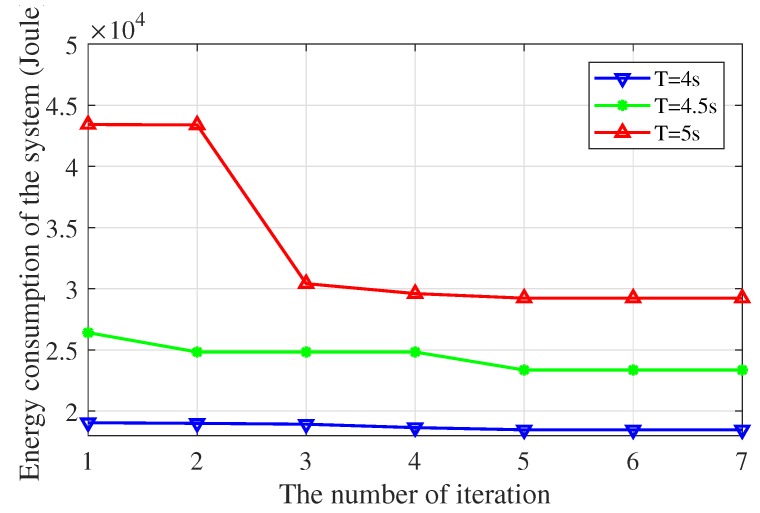
The total energy consumption of the system ($I_{1}=4\times 10^{7}$ bits, $I_{2}=5\times 10^{7}$ bits, $I_{3}=6\times 10^{7}$ bits, $I_{4}=7\times 10^{7}$ bits and $I_{5}=8\times 10^{7}$ bits).

**Figure 10 sensors-19-04521-f010:**
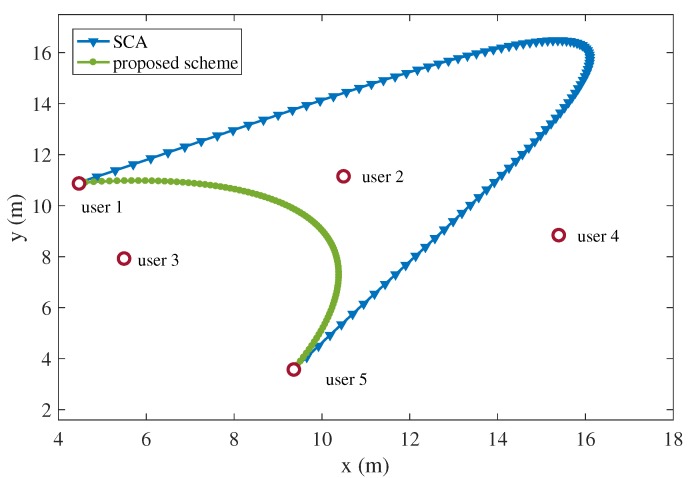
The trajectory of the UAV using the proposed scheme and the SCA scheme under the time constraint T=5 s ($I_{1}=I_{3}=I_{5}=2\times 10^{7}$ bits, $I_{2}=I_{4}=8\times 10^{7}$ bits).

**Figure 11 sensors-19-04521-f011:**
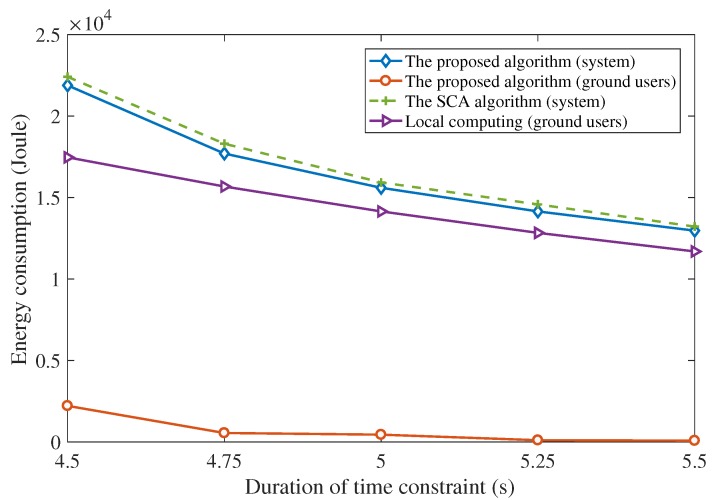
The total energy consumption of the MEC system under different time constraints ($I_{1}=I_{3}=I_{5}=2\times 10^{7}$ bits, $I_{2}=I_{4}=8\times 10^{7}$ bits).

**Table 1 sensors-19-04521-t001:** List of symbols.

Symbol	Description
K,k	The set of ground users, k∈K
*K*	Number of ground users
$A_{k}$	The notation of user *k*’s task
$I_{k}$	Task input-data size of user *k*
$\tau_{k}$	Task deadline of user *k*
$C_{k}$	Number of CPU cycles needed to compute one input bit of user *k*
$O_{k}$	Ratio of the number of output bits to the number of input bits for user *k*
*T*	Time constraint of the tasks
*N*	Number of time slots of *T*
Δ	Time duration of one slot
δ	Time duration of one sub-slot
$q_{k}$	Location of user *k*
$q_{u}\left[n\right]$	Location of the UAV in *n*th slot
v[n]	Velocity of the UAV in *n*th slot
$V_{max}$	Maximal velocity of the UAV
*h*	Variable of the UAV’ altitude
*H*	Fixed altitude of the UAV
*B*	Communication bandwidth
$\sigma^{2}$	Noise power at the receiver
$g_{0}$	Channel power gain at reference distance 1m
$g_{k}\left[n\right]$	Channel gain from the ground user *k* to the UAV in *n*th slot
$I_{k}^{u}\left[n\right]$	Number of uploading bits of user *k* in *n*th slot
$I_{k}^{c}\left[n\right]$	Number of computing bits of the UAV for user *k* in *n*th slot
$I_{k}^{d}\left[n\right]$	Number of downloading bits from the UAV to user *k* in *n*th slot
$f_{U,k}\left[n\right]$	Frequency of the UAV’s CPU in the *n*th slot for computing the tasks of user *k*
$f_{u}\left[n\right]$	Frequency of the UAV’s CPU in the *n*th slot
$E_{U,k}^{c}\left[n\right]$	Computation energy consumption of the UAV for ground user *k* in the *n*th slot
$\gamma_{u}$	Effective switched capacitance of the UAV’s CPU
κ	Coefficient of the flying energy consumption (κ=0.5MΔ)
*E*	Total energy consumption of the UAV-enabled MEC system
$E_{G}$	Energy consumption of *K* ground users
$E_{U}$	Energy consumption of the UAV
$E_{G}^{U}$	Communication energy consumption for uploading data of the ground users
$E_{U}^{C}$	Computing energy consumption of the UAV
$E_{U}^{F}$	Flying energy consumption of the UAV
$E_{U}^{D}$	Energy consumption of downloading the results from the UAV to ground users
$I_{k}^{u*}\left[n\right]$, $I_{k}^{c*}\left[n\right]$, $I_{k}^{d*}\left[n\right]$	Optimal number of $I_{k}^{u}\left[n\right]$, $I_{k}^{c}\left[n\right]$ and $I_{k}^{d}\left[n\right]$ under the given trajectory of the UAV
λ, μ, ν, ρ, β, η	Dual variables according to the constraints (21b)–(21k)
$\lambda_{j}$, $\mu_{k,j}$, $\nu_{k,j}$, $\rho_{k,j}$, $\beta_{k,n,j}$, $\eta_{k,n,j}$	Dual variables at the *j*th iteration in the subgradient method
$\alpha_{j}^{\left(\lambda \right)}$, $\alpha_{j}^{\left(\mu \right)}$, $\alpha_{j}^{\left(\nu \right)}$, $\alpha_{j}^{\left(\rho \right)}$, $\alpha_{j}^{\left(\beta \right)}$, $\alpha_{j}^{\left(\eta \right)}$	*j*th step size computed in subgradient algorithm
$g_{j}^{\left(\lambda \right)}$, $g_{j}^{\left(\mu \right)}$, $g_{j}^{\left(\nu \right)}$, $g_{j}^{\left(\rho \right)}$, $g_{n,j}^{\left(\beta \right)}$, $g_{n,j}^{\left(\eta \right)}$	Subgradients calculated by ([Disp-formula FD25a-sensors-19-04521])–(25f)

**Table 2 sensors-19-04521-t002:** List of symbols.

Parameter	Description	Value
$C_{k}$	Number of CPU cycles needed to compute one input bit of user *k*	1500 cycles/bit
$O_{k}$	Ratio of the number of output bits to the number of input bits for user *k*	0.5
*K*	Number of ground users	5
*N*	Number of time slots of *T*	100
$V_{max}$	Maximal velocity of the UAV	15 m/s
*H*	Altitude of the UAV	10 m
*B*	Communication bandwidth	40 Mhz
$\sigma^{2}$	Noise power at the receiver	$10^{-9}$ W
$g_{0}$	Channel gain from the ground user *k* in *n*th slot	−30 dB
$\gamma_{u}$	Effective switched capacitance of the UAV’s CPU	$10^{-28}$
κ	Coefficient of the flying energy consumption (κ=0.5MΔ)	0.0675
*E*	Total energy consumption of energy consumption of the UAV-enabled MEC system	$5\times 10^{5}$
$\xi_{1}$, ξ	tolerant thresholds of the iterations	$5\times 10^{-4}$

**Table 3 sensors-19-04521-t003:** Run time of Algorithm 1.

(N,K)	(25,2)	(50,2)	(75,2)	(25,4)	(50,4)	(75,4)	(25,6)	(50,6)	(75,6)
Algorithm 1	23.56 s	41.31 s	108.42 s	25.48 s	81.09 s	203.32 s	71.73 s	132.18 s	315.42 s

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
