# Peer review of "Energy-Efficient UAV-Enabled MEC System: Bits Allocation Optimization and Trajectory Design"

_sensors, 2019, doi:10.3390/s19204521_

Round 1

Reviewer 1 Report

The paper presents an algorithm for offloading mobile computing into passing UAV. The presentation is clear and the analysis is sound. I would like the authors to discuss more about energy consumption during communication. It is well known that distance greatly affect wireless communication, but the presented approach seems rather simplistic in this regard. However, being such a complex problem, I do not believed that this is a major issues, just a future improvement.

My main concerns is whether this kind of computational offloading is applicable, as I can think only in some scenario. However, I think that the works has its merits, and that nowadays it could be used in some scenarios, such as military ones.

Reviewer 2 Report

The authors study the problem of data offloading in a mobile edge computing environment assisted by unmanned aerial vehicles. The topic of the paper The topic of the paper is of great interest as the unmanned aerial vehicles, as well as the mobile edge computing environment become a need in future communication and especially within the Internet of Things era, where billions of low energy devices communicate among each other, are connected to the network, and ask for computing support. Overall, the paper is well written and easy to follow. The proposed analysis is concrete and correct and the paper has an easy-to-follow structure. One drawback that the authors should take care of is the motivation of their research and the way they position their research work within the recent advances in the unmanned aerial vehicles era. Specifically, in the Introduction, the authors should provide a section discussing the problems of resource management in the unmanned aerial vehicles environments, especially, in the uplink communication that is considered in this paper, e.g., Wireless powered Public Safety IoT: A UAV-assisted adaptive-learning approach towards energy efficiency. Journal of Network and Computer Applications, 123, 69-79, 2018, Self-Adaptive Energy Efficient Operation in UAV-Assisted Public Safety Networks. In 2018 IEEE 19th International Workshop on Signal Processing Advances in Wireless Communications (SPAWC) (pp. 1-5). IEEE, Regret based learning for uav assisted lte-u/wifi public safety networks, in: Global Communications Conference (GLOBECOM), IEEE, 2016, pp. 1–7, Unmanned aerial vehicle with underlaid device-to-device communications: Performance and tradeoffs, IEEE Transactions on Wireless Communications 15 (6) (2016) 3949–3963.  The authors should update the introduction and the references list accordingly to better present the recent resource management frameworks in unmanned aerial vehicles environments, where the following mobile edge computing setup is based on. In Table 1, the authors should include the units of all the variables and parameters considered in this paper.  Moreover, the authors should present the complexity analysis of the algorithm 1 and present some numerical results to justify the easy implementation of the algorithm, as it is feasibility in realistic implementations. Additionally, figures 6 and 7 should be enlarged. Overall, this is a very interesting paper that should be slightly improved in order to be better motivated and positioned within the existing literature. The manuscript needs a detailed check for minor typos, grammar and syntax errors, e.g., Keywords: … design.).

Reviewer 3 Report

The topic covered by this manuscript is interesting. I recommend a major revision of this paper. Some comments should be addressed as the following:

Please put a clear introduction of feasible applications and scenarios that can utilize the advantages of the UAV-enabled MEC system.  Even though we can find the abbreviations in Table 1, each abbreviation should be defined for the first time it is used in the paper. Computing energy is not always true following Equation (5). Because each task has a different complexity, computing energy must consider the complexity of the task. For your reference: (1) Dynamic computation offloading for mobile-edge computing with energy harvesting devices, IEEE JSAC, 2016; (2) SGCO: Stabilized green crosshaul orchestration for dense IoT offloading services, IEEE JSAC, 2018; etc. Equation (20a), because a task was uploaded in n-th slot might be downloaded in the next slots (e.g., n+3 th slot), it is not true if considering both I_n^u[n] and I_n^d[n] in one equation. Please analyze time and space complexity of the proposed algorithm.

Reviewer 4 Report

The paper proposes an unmanned aerial vehicle (UAV) enabled mobile edge computing (MEC) system where the total energy consumption is minimized. The paper is interesting and generally well written. However, some issues need to be solved:

There are two assumptions that the authors seem to consider in their paper which are not explicitly presented: a) the authors considered that the mobile devices are always in the communication range of UAV; b) the UAV is considered to be a rotary wing UAV (a fixed-wing UAV cannot stand still  - see Figure 5 last rectangle); Moreover the UAV’s flight dynamic is simplified (a rotary-wing UAV has small altitude changes when changing its direction – see for example the squared trajectory); The assumption I described at 1a) is unrealistic for networks of mobile devices distributed over a large area. In this case, some details about the scalability of the algorithm are needed. In my perspective, the trajectories presented in Figure 4 need to be changed such that the original/starting point to be also the finishing point (to be ready for a new set of transmissions). In this case, probably, the fixed point will be the best choice. In Figure 4, the distance between the mobile devices has to be enlarged (why use a UAV for mobile devices placed in a circle of 7.5 m radius?); Some details about the relation between the UAV speed and T or delta may be useful; Lines 116-118: the Roman numerals must be changed into Arabic numerals; Line 17: there is an unwanted “)” there.

Round 2

Reviewer 2 Report

The authors have successfully addressed the reviewer's comments. The manuscript has been substantially improved and it is ready for publication.

Author Response

Dear Reviewer:
Thank you very much for your supervision of the reviewing process of our Manuscript ID: sensors-605073 entitled " Energy-Efficient UAV-Enabled MEC System: Bits Allocation Optimization and Trajectory Design". We also highly appreciate the Reviewer's carefulness, conscientious, and the broad knowledge on the relevant research fields, since you have given us a number of beneficial and constructive comments and suggestions. 

Once again, thank you very much for your comments and suggestions. The valuable suggestions and comments are instructive and meaningful
to our future research!

Reviewer 3 Report

Thank you for your hard work to revise the manuscript. I agreed with your responses. Regarding time complexity, it is quite high when N increases, I recommend the authors to put it as one of the future works in your paper to reduce it. I suggest an acceptance after the authors do this minor comment.

Reviewer 4 Report

The authors have successfully solved all the issues I raised.

Author Response

(The authors gave the same response as above.)
